# Effects of Mini-Basketball Training Program on Executive Functions and Core Symptoms among Preschool Children with Autism Spectrum Disorders

**DOI:** 10.3390/brainsci10050263

**Published:** 2020-04-30

**Authors:** Jin-Gui Wang, Ke-Long Cai, Zhi-Mei Liu, Fabian Herold, Liye Zou, Li-Na Zhu, Xuan Xiong, Ai-Guo Chen

**Affiliations:** 1College of Physical Education, Yangzhou University, Yangzhou 225127, China; jinguiwang5715@163.com (J.-G.W.); kelongcai@163.com (K.-L.C.); 15161885591@sohu.com (Z.-M.L.); movmu7@sina.com (X.X.); 2Research Group Neuroprotection, German Center for Neurodegenerative Diseases (DZNE), Leipziger Str. 44, 39120 Magdeburg, Germany; Fabian.herold@dzne.de; 3Department of Neurology, Medical Faculty, Otto von Guericke University, Leipziger Str. 44, 39120 Magdeburg, Germany; 4Exercise and Mental Health Laboratory, School of Psychology, Shenzhen University, Shenzhen 518060, China; liyezou123@gmail.com; 5School of Physical Education and Sports Science, Beijing Normal University, Beijing 100000, China; zhulina827@163.com

**Keywords:** exercise, executive functions, core symptoms, children, autism spectrum disorders

## Abstract

This study examined the effects of a 12-week mini-basketball training program (MBTP) on executive functions and core symptoms among preschoolers with autism spectrum disorder (ASD). In this quasi-experimental pilot study, 33 ASD preschoolers who received their conventional rehabilitation program were assigned to either a MBTP group (*n* = 18) or control group (*n* = 15). Specifically, the experimental group was required to take an additional 12-week MBTP (five days per week, one session per day, and forty minutes per session), while the control group was instructed to maintain their daily activities. Executive functions and core symptoms (social communication impairment and repetitive behavior) were evaluated by the Childhood Executive Functioning Inventory (CHEXI), the Social Responsiveness Scale-Second Edition (SRS-2), and the Repetitive Behavior Scale-Revised (RBS-R), respectively. After the 12-week intervention period, the MBTP group exhibited significantly better performances in working memory (*F* = 7.51, *p* < 0.01, partial *η*^2^ = 0.195) and regulation (*F* = 4.23, *p* < 0.05, partial *η*^2^ = 0.12) as compared to the control group. Moreover, the MBTP significantly improved core symptoms of ASD preschoolers, including the social communication impairment (*F* = 6.02, *p* < 0.05, partial *η*^2^ = 0.020) and repetitive behavior (*F* = 5.79, *p* < 0.05, partial *η*^2^ = 0.016). Based on our findings, we concluded that the 12-week MBTP may improve executive functions and core symptoms in preschoolers with ASD, and we provide new evidence that regular physical exercise in the form of a MBTP is a promising alternative to treat ASD.

## 1. Introduction

Autism spectrum disorder (ASD) is a neurodevelopmental disorder with several core symptoms (social skill deficits, communication problems, stereotyped and repetitive behavior) [1]. Empirical evidence suggests that these problem behaviors do not merely involve the typical clinical manifestations, but also limit opportunities for achieving health benefits, academic achievement, and social integration [2]. Recently, accumulating evidence has indicated that children with ASD have also presented with executive dysfunction [3].

Executive functions are brain-based skills required to successfully carry out goal-directed behaviors, with three primary domains that are generally categorized: (1) inhibition is the ability to voluntarily inhibit impulsive responses, (2) regulation involves the mental ability to selectively shift attention between two tasks, and (3) working memory is the ability to hold the meaningful information for decision making, planning, and organization [4]. In the context of ASD, executive functions are proposed to significantly associate with specific impairments (e.g., theory of mind, social cognition, social impairment, restricted and repetitive behaviors, and quality of life) [5,6,7,8]. It is also worth mentioning that the prefrontal cortex as a key structure is extensively investigated to explore neural mechanisms related to executive functions [9]. In line with this, Hill’s executive dysfunction theory has proposed that in individuals with ASD, difficulties in initiating new non-routine actions and stereotypical behaviors are linked to frontal lobe dysfunction [10]. According to the above-mentioned evidence concerning social and behavioral deficits as well as executive dysfunction, it is very likely that cognitive deficits in ASD children may persist into adulthood.

Currently, there is no available pharmacological therapy for treating ASD children [11,12]. Hence, ASD children are usually treated by behavioral therapies, but it is difficult for many families to access the time-consuming counseling sessions or to implement long-term and high-quality behavioral interventions in their daily routines. Moreover, those behavioral treatments are relatively expensive and challenging, which may not be affordable for all ASD families. More unfortunately, the conduction of behavioral therapies does not produce significant positive changes in the underlying deficits that promote the behavioral manifestations of ASD in children. To overcome the limitations of these existing behavioral interventions, it is urgently needed to search for effective and alternative therapeutic strategies (like exercise training) for treatment of ASD children [13]. 

Accumulating evidence has shown that physical exercise interventions could be a valuable alternative to the existing behavioral therapies, as it was observed that acute physical exercises can improve cognitive functions, at least transiently, in ASD children [14,15,16,17,18,19]. Likewise, previous studies also showed that long-term physical exercise interventions had positive effects on executive functions among school-aged children who were diagnosed with ASD [13,20,21,22,23]. Moreover, it is reported that in children with ASD, the rehabilitative effects of physical exercise interventions (e.g., exergaming and Karate techniques training) on social communication, repetitive behaviors, and cognitive performance can be maintained, at least, up to one month after the cessation of the intervention [22]. 

Notably, previous studies on physical exercise intervention focused on school-aged children with ASD, rather than preschoolers. As it is well-known that preschool age is a critical period for dramatic growth and the development of executive functions [24], thus it would also be appropriate to start intervention programs during this developmental stage. This assumption is supported by the growing evidence suggesting that early intensive interventions would ameliorate symptoms of ASD children, which was associated with an increased likelihood of moving into socialized environments in later life periods [25,26]. Collectively, physical exercise interventions that start at a preschool age stage seem to also be beneficial for this special population. Hence, to fill this knowledge, we aimed to investigate the effects of an exercise intervention program on executive function and core symptoms of ASD in a sample of 3–6-year-old preschoolers with ASD. As it has been speculated [27,28,29] and demonstrated [30] that cognitively and/or coordinatively demanding physical exercises are more efficient to improve executive functioning in children than purely aerobic exercises, we implemented a mini-basketball training program (MBTP) in this intervention study. Mini-basketball is a cognitively challenging and coordination training program which trains speed and strength, as well as social and cognitive skills [31,32]. Based on the promising evidence which suggests that physical exercise can improve social skills and cognitive performance, we hypothesized that the 12-week MBTP would result in improvements of executive functioning and core symptoms in preschoolers with ASD as compared to the control group.

## 2. Materials and Methods

### 2.1. Study Design

The study was a quasi-experimental design in which “between-subject factor” was “group/condition” and “within-subject factor” is “time”. This study was conducted between October and December 2018 in Yangzhou, China, with ethics approval from the Ethics and Human Protection Committee of the Affiliated Hospital of Yangzhou University. Study protocol was registered with the Chinese Clinical Trial Registry (ChiCTR1900024973) before initiating our experiments and all study procedures are in accordance with the latest version of the Declaration of Helsinki. Written informed consent were obtained from parents of ASD children.

### 2.2. Participants

Children aged 3–6 years, meeting the Diagnostic and Statistical Manual of Mental Disorders, 5th edition, criteria for ASD [1], were recruited from Yangzhou Chuying Child Development Center and Starssailor Education Institute (Yangzhou, P. R. China). Children were not eligible if they met one of the following exclusion criteria: (1) received basketball training or regular participation in physical exercise in the past 6 months, (2) one or more co-morbid psychiatric disorders, (3) a neurological disorder (e.g., epilepsy, phenylketonuria, fragile X syndrome, tuberous sclerosis), (4) visual and auditory disorders, (5) a medical history of head trauma, and (6) any medical condition that does not allow exercise participation (e.g., heart disease, operations or fractures within the last six months).

Ninety-four children who met the diagnosis for ASD were initially screened in the two institutes. Of the initially diagnosed ASD preschoolers, 35 children who did not meet study criteria or declined to participate in this study were excluded so that 59 participants were finally eligible. Given that this is a pilot study, a quasi-experimental design was adopted in which eligible participants were geographically arranged into two groups: (1) participants from the Chuying Child Development Center were considered as the control group (*n* = 29), and (2) participants from the Starssailor Education Institution were chosen as the MBTP group (*n* = 30). Notably, only 33 participants (28 boys and 5 girls; 4.924 ± 0.697 years) were finally included for data analysis (Figure 1) after removal of 26 participants due to (1) injury that was not related to basketball (*n* = 1); (2) sickness (*n* = 4); (3) schedule conflict of parents (*n* = 6); (4) and unwillingness to participate in the functional magnetic resonance imaging (fMRI) task (*n* = 15). 

### 2.3. Power Calculation

Prior sample size was calculated using G*Power [33] while we selected key parameters: (1) a medium effect size (Cohen’s f = 0.25), (2) power of 0.80, (3) alpha of 0.05, and (4) repeated measures analysis of variance (ANOVA) with the two groups (MBTP and control). This indicated that a sample of twenty-four (twelve participants in each of the two groups) can generate a statistical significance. Given that ASD children might drop out over a 12-week intervention period, we increased this by 37.5%, which has yielded the final sample of thirty-three that should be included in this study.

### 2.4. Mini-Basketball Training Program

In this present study, the mini-basketball training program (MBTP) was adopted from previous studies [31,32,34]. This exercise program has been shown to be a safe, enjoyable, readily accessible, and easy-to-administer program, which is popular among preschool children and their parents. This program with various levels of difficulty is appropriately designed for this special group and ASD children were taught in a progressive manner (three stages are detailed in Appendix A). As they progressed, cognitively demanding game play was arranged to train the executive functions of ASD children. Each training session was carried out in collective classes, which could facilitate social interaction and communication among the participating children with ASD [35]. Parents of the participating children were encouraged to join each training session, which could create a more enjoyable and positive climate, leading to more effective social interactions among all participants. On the other hand, if their parents were present, it potentially makes ASD children feel more comfortable so that they are more likely to succeed in motor skill learning. As recommended by the American College of Sports Medicine, moderate exercise intensity was set, with 60–69% of maximum heart rate (MHR; MHR was determined by the formula: MHR = 220 − age of the participant) [36]. Exercise intensity was monitored using heart rate monitors (POLAR M430) throughout all sessions of the experiment.

In this pilot study, the 12-week MBTP was arranged for ASD children in the experimental group. Weekly five sessions (60 sessions = 12 weeks × 5 sessions) were carried out, with each session lasting 40 min. Specifically, each session included four stages of (a) 5-min warm-up, (b) 20-min basic basketball skill learning, (c) 10-min basketball games, and (d) 5-min cool-down. Parts (b) and (c) involved 30 min of moderate-intensity physical activity, indicated by 129–149 heart beats per minute on average. 

### 2.5. Measurements

Demographic information (age, gender, weight and height) were obtained at baseline and are presented in Table 1. Severity of participant was assessed using the Childhood Autism Rating Scale (CARS) [37] and clinical assessment report. Additionally, previous studies have indicated that core symptoms were associated with sleep disorders and eating behaviors in children with ASD [38,39]. To control for these potential confounding variables, sleep problems and eating behavior were collected at baseline assessment. First, sleep problems were assessed using the Children’s Sleep Habits Questionnaire (CSHQ) [40], administered by their parents. Second, eating style was assessed by a parent-report Child Eating Behavior Questionnaire (CEBQ) [41]. Executive functions as primary outcomes and core symptoms as secondary outcomes were also measured in this pilot study. Their assessment tools are discussed below. 

#### 2.5.1. Assessment of Executive Functions

In this pilot study, the validated Childhood Executive Functioning Inventory (CHEXI) in Chinese was used to measure severity of executive functions [42]. The revised version consisted of 24 items (1 = completely inconsistent to 5 = completely consistent) within three dimensions (regulation, inhibition, and working memory). A sum score of 120 can be obtained, with higher scores indicating worse executive function [43].

#### 2.5.2. Assessment of Core Symptoms

Social Communication Impairment was measured using the Social Responsiveness Scale Second Edition (SRS-2) [44]. It consisted of 65 items, with each item rated on a 4-point scale ranging from 1 (not true) to 4 (almost always true). It can generate a total score of 75, with higher scores indicating more severe impairment in social communication. Likewise, parent-reported behavioral outcomes were also collected with the validated assessment tool [44]. 

Repetitive behaviors were measured using the Repetitive Behavior Scale-Revised (RBS-R) [45]. This informant-based questionnaire was designed to provide a quantitative, continuous measure of the full spectrum of various repetitive behaviors among individuals with ASD. In the RSB-R, 43 items are grouped conceptually. Each item is rated on a 4-point Likert scale for severity (0 = behavior does not occur, 1 = behavior occurs and is a mild problem, 2 = behavior occurs and is a moderate problem, 3 = behavior occurs and is a severe problem). Higher scores indicate more severe behaviors. 

### 2.6. Procedure

All potential participants were screened by physicians in the Yangzhou Maternal and Child Care and Service Centre. Meanwhile, written informed consent was obtained from mothers of all participating children after the experimental procedures had been fully explained. Eligible ASD children were assigned to either a MBTP group or control group. During the first visit, their legal guardians completed all paperwork, including the CHEXI, SRS-2, RBS-R, CSHQ, and CEBQ. Both MBTP and control groups had received the similar conventional rehabilitation program. Notably, only ASD preschoolers in the experimental group were arranged to attend the MBTP, whereas the control group maintained their unaltered lifestyle. The intervention duration of the MBTP was 12 weeks in total. Each session was conducted by two certified physical educators. For the safety consideration and teaching effectiveness, at least one parent of the participating children was required to join the class and play with their child throughout the entire training session. The training program was implemented as described previously (see Section 2.4 Mini-Basketball Training Program). ASD children who completed this intervention and all assessments received a basketball as fair remuneration

### 2.7. Statistical Analysis

A two-way repeated design was employed with group and time as independent variables. To ensure homogeneity, significant group differences on potential confounding variables (age, gender, eating behaviors, and sleep behaviors) were tested using a *t*-test or a χ^2^-test. Then, a mixed ANOVA was employed to determine significant group differences on primary and secondary outcomes between baseline and post-intervention. If significant interaction effects existed, simple effects were performed. Effect sizes (Cohen’s d) that reflect the magnitude of the intervention effect were calculated and partial eta-squared (η^2^) values were reported for significant main effects and interactions. Data was presented as descriptive statistics: mean ± standard deviation (M ± SD). An α of 0.05 was used as the level of statistical significance for all statistical analyses, which were conducted using jamovi 1.0.7 (Retrieved from https://www.jamovi.org).

## 3. Results

### 3.1. Participant Characteristics

There was no significant group difference on gender (chi-square = 0.07, *p* > 0.05), age (*t*(1,31) = 1.74, *p >* 0.05), height (*t*(1,31) = 1.96, *p* > 0.05), weight (*t*(1,31) =1.64, *p* > 0.05), severity (CARS) (*t*(1,31) = 1.08, *p* > 0.05), physical fitness (*t*(1,31) = 1.59, *p* > 0.05), sleep problems (CSHQ) (*t*(1,31) = −0.92, *p >* 0.05), and eating behavior (CEBQ) (*t*(1,31) = −0.03, *p >* 0.05). Results indicated homogeneity between the two groups. The demographic characteristics of participants in both groups are summarized in Table 1.

### 3.2. Executive Functions

A significant group × time interaction (*F*(1,31) = 7.51, *p* < 0.05, partial *η*^2^ = 0.195) was found on working memory (Table 2). Results from the follow-up analysis have shown no significant group differences on baseline scores between the MBTP and control groups. Notably, significantly lower scores in the MBTP were observed at the post-intervention as compared to the baseline (*t*(1,32) = 1.57, *p* < 0.05, *d* = 0.274), but not in the control group.

A significant group × time interaction (*F*(1,31) = 5.66, *p* < 0.05, partial *η*^2^ = 0.08) was found on inhibition (Table 2). Results from the follow-up analysis have shown no significant group differences at the baseline assessment. Notably, significantly higher scores in the control group were observed at the post-intervention as compared to the baseline (*t*(1,32) = −1.84, *p* < 0.05, *d* = 0.320), but not in the MBTP group. 

A significant group × time interaction (*F*(1,31) = 4.23, *p* < 0.05, partial *η*^2^ = 0.12) was found on regulation (Table 2). Results from the follow-up analysis have shown no significant group differences at the baseline. Notably, significantly lower scores in the MBTP were observed at the post-intervention as compared to the baseline (*t*(1,32) = −2.06, *p* < 0.05, *d* = 0.359), but not in the control group (Figure 2).

### 3.3. Core Symptoms

For social communication, a statistically significant group × time interaction was present (*F*(1,31) = 6.02, *p* < 0.05, partial *η*^2^ = 0.020) (Table 2). Results from the follow-up analysis have shown no significant group differences at the baseline. Notably, significantly lower scores in the MBTP were observed at the post-intervention, as compared to the baseline (*t*(1,32) = 3.04, *p* < 0.05, *d* = 0.528), but not in the control group. 

For repetitive behavior, a significant statistical difference between the groups (*F*(1,31) = 5.93, *p* < 0.05, partial *η*^2^ = 0.144) was found. A statistically significant group × time interaction was present (*F*(1,31) = 5.79, *p* < 0.05, partial *η*^2^ = 0.016) (Table 2). Results from the follow-up analysis have shown no significant group differences at the baseline. Notably, significantly lower scores in the MBTP were observed at the post-intervention as compared to the baseline (*t*(1,32) = 2.57, *p* < 0.05, *d* = 0.447), but not in the control group. 

For the primary and secondary outcomes, means and standard deviations at the baseline and post-intervention are presented in Table 2. The follow-up contrasts are show in Figure 2 and Figure 3.

## 4. Discussion

The present study investigated the effects of a 12-week MBTP on executive functions and core symptoms among preschoolers with ASD. With the continual employment of fundamental movement skill training and basketball games in MBTP, the main findings revealed that MBTP improved executive function, social communication, and repetitive behavior. Potential explanations for our results are discussed below. 

### 4.1. Executive Functions

All aspects of executive functions improved following the 12-week intervention, including working memory, inhibition, and regulation. Recent studies in children with and without disabilities showed that fundamental motor training (e.g., speed-agility, strength training, and coordinating upper- and lower-extremity movements) and complex motor skills training could benefit cognitive performance, which is consistent with our findings [20,21,35,46,47,48,49]. Furthermore, our results are in line with previous studies investigating the beneficial effects of cognitive-motor exercise training for executive function [27,28,29,30,50]. The superior effects of cognitively engaging exercise training might be related to the neurocognitive demands which are posed to the brain in order to execute the motor-cognitive tasks [51]. In the literature, this effect has also been referred to as the “guidance effect” [52,53,54]. In particular, MBTP includes the learning of new and complex motor skills (inter-limb coordination), which demands various cognitive domains during the game playing part of the exercise sessions. The cognitive demands posed by the MBTP might train more or less indirectly specific aspects of executive functions (e.g., inhibition or working memory). For instance, the basketball game play requires working memory to update and store the new and old information alternately (position of the players on the field). However, further investigations are needed to provide empirical support for our assumptions.

Another possible explanation for cognitive benefits is the “facilitation effect”. It has emphasized the pronounced release of neurochemicals such as brain-derived neurotrophic factor (BDNF) in response to physical exercise and physical training [52,53,54]. In this regard, it was observed that after 12 weeks of endurance training, the resting concentration of the serum level of BDNF and the working memory performance increased in a sample of healthy adolescent subjects [55]. Whether such changes in serum or plasma levels of BDNF occur after the MBTP in preschoolers with ASD and how these changes might positively influence cognitive performance would be an interesting topic for further studies in this field. In addition, neural correlates (e.g., changes in functional brain activity patterns) of the observed cognitive improvements need to be further investigated. Speculatively, MBTP might positively influence functional brain activation since previous studies reported that in overweight children, significant functional brain changes had occurred after an eight-month aerobic training intervention [56,57].

### 4.2. Core Symptoms

The findings of the current study support our hypothesis that a 12-week MBTP improved social communication and repetitive behavior among preschoolers with ASD. These findings were similar to previous studies suggesting that social behavior improved after prolonged training in horse riding [58], aquatic group exercise [59], or movement skills [22]. In addition, martial arts training (e.g., techniques training) for children with ASD effectively reduced their repetitive behavior [60]. Such observed improvements in social communication may be attributed to that ASD children who were repeatedly exposed to a rich environment had the opportunity to improve imitation skills, potentially leading to better social skills. This assumption is buttressed by one study which reported that imitation treatment was an effective approach to improve social communication in ASD children [61]. An alternative exploratory approach is that MBTP was implemented in a positive and collaborative environment (individual games, collaborative games, and competitive games) which, in turn, may promote their social and emotional health. Besides, MBTP contains many repeated body movements that are similar to repetitive behavior, which may contribute to physical exhaustion so that frequency of stereotypic behaviors were reduced in this study. 

Leung et al. had found that executive dysfunction is often predictive for the core symptoms of children with ASD. In other words, children with superior executive functions were more likely to have lower core symptoms of ASD [6]. Given that the time from birth to 8 years is a critical period in the brain development, starting interventions in early age stages (preschool age in the current study) is probably more beneficial for the development of executive functioning in children with ASD. Likewise, our results suggest that preschoolers with ASD who participated in a 12-week MBTP have shown improved executive functions and reduced core symptoms of ASD. Hence, our promising findings support the idea that physical exercise interventions such as MBTP can be considered as a therapeutic tool for treating ASD preschoolers. 

In summary, the findings of the present study suggest that the 12-week MBTP effectively improved executive function and core symptoms in preschoolers with ASD. Thus, this study provides initial evidence that MBTP serves as an effective, inexpensive, easily accessible intervention that can be applied in various settings across different cultures and countries. MBTP is a promising alternative intervention program that meets the need for a globally applicable intervention model for ASD preschoolers.

### 4.3. Strengths and Limitations

A clear strength of our study is the application of an alternative approach to treat ASD preschoolers. Specifically, we implemented a multidimensional physical exercise program, namely a mini-basketball training program that did not simply include the training of basic movement skills, but were also designed to promote behavior reinforcement. Hence, MBTP is different from previous training programs that were applied to treat preschool children with ASD. A further strength of this study is the rigorous control for several potential confounders, which could strengthen the assumption that the observed training-related gains in executive function and social communication can be attributed to the conduction of the MBTP. Nevertheless, it is important to emphasize that there are still several limitations in the current study which need further discussion. Firstly, we used a quasi-experimental design and thus the participants were not randomly assigned. On the other hand, since baseline scores were not significantly different between the two groups, this non-random assignment may not affect our findings. Secondly, although CHEXI, SRS-2, and RBS-R were validated scales to specifically assess executive functions and core symptoms, they are subjective measures, administered by parents of the participating children. It must be admitted that there is currently no other tool available which would allow for a more objective assessment of social communication impairments of individuals with ASD. Thus, positive effects on social communication in this study should be interpreted cautiously. Finally, except for age and gender, other demographic information including IQ, language level, and severity of symptoms were not collected in this pilot study. Future studies on this topic should include these variables. 

## 5. Conclusions

The present study provides initial evidence that 12 weeks of MBTP may positively influence executive functions and core symptoms in preschoolers with ASD. These promising findings suggest that a mini-basketball training program can be used as a complementary intervention to alleviate core symptoms of ASD and to enhance executive functioning in preschoolers with ASD. Further research is needed to investigate the underlying neurobiological processes of the observed behavioral improvements.

## Figures and Tables

**Figure 1 brainsci-10-00263-f001:**
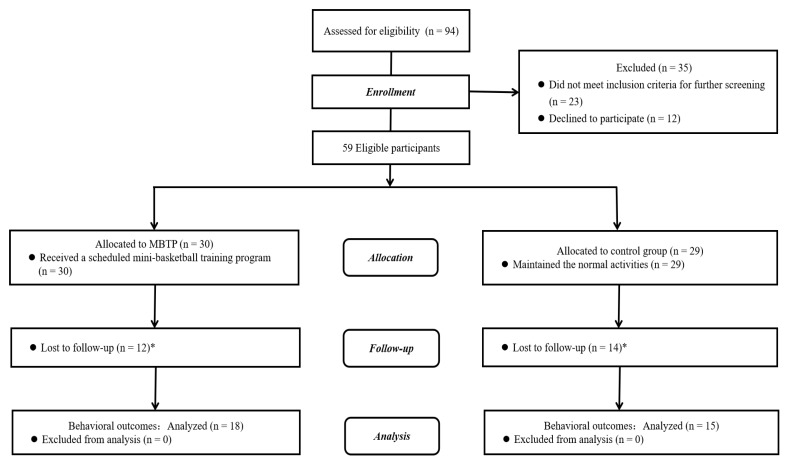
Participant flow chart. * Twenty-six children’s parents did not finish the assessment in the post-test.

**Figure 2 brainsci-10-00263-f002:**
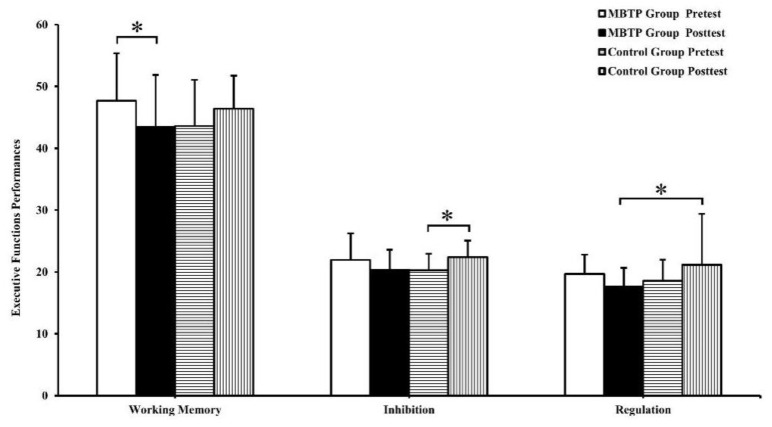
Performances for executive functions (mean and standard deviation) of time point (baseline versus post-test) and group (MBTP versus Control), * *p* < 0.05.

**Figure 3 brainsci-10-00263-f003:**
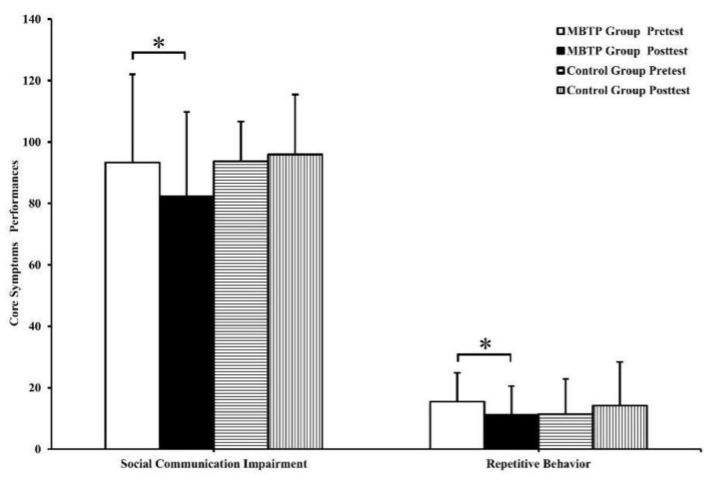
Performances for core symptoms (mean and standard deviation) of time point (baseline versus post-test) and group (MBTP versus control), * *p* < 0.05.

**Table 1 brainsci-10-00263-t001:** Baseline characteristics of participants (M ± standard deviation (SD)).

Measure	MBTP Group	Control Group	*p*
*N*	18	15	-
Gender (boys/girls)	15/3	13/2	0.790
Age(years)	5.11 ± 0.65	4.70 ± 0.70	0.092
Body height (cm)	113.94 ± 8.49	108.93 ± 5.57	0.059
Body mass (kg)	20.63 ± 3.32	18.89 ± 2.90	0.112
CARS	45.94 ± 21.37	39.80 ± 5.24	0.287
Physical Fitness	20.39 ± 2.87	18.73 ± 3.15	0.125
CSHQ	55.72 ± 4.73	58.60 ± 12.29	0.366
CEBQ	54.22 ± 8.88	54.40 ± 20.05	0.973

MBTP: mini-basketball training program; CARS: Childhood Autism Rating Scale; CSHQ: The Children’s Sleep Habits Questionnaire; CEBQ: Children’s Eating Behavior Questionnaire. *p-*Values are calculated using independent samples *t-*tests for continuous variables and chi-square tests for categorical variables between groups.

**Table 2 brainsci-10-00263-t002:** Analysis of MBTP and Control groups for executive functions and core symptoms variables (M ± SD).

	MBTP Group (*n* = 18)		Control Group (*n* = 15)	
Variable	Baseline	Posttest	Baseline	Posttest
Executive Functions				
Working memory	47.67 ± 7.71	43.44 ± 8.42	43.60 ± 7.47	46.40 ± 5.36
Inhibition	21.94 ± 4.29	20.33 ± 3.25	20.27 ± 2.66	22.40 ± 2.67
Regulation	19.67 ± 3.11	17.67 ± 2.99	18.53 ± 3.42	21.13 ± 8.25
Core Symptoms				
SRS-2	93.28 ± 28.75	82.22 ± 27.55	93.72 ± 12.92	95.93 ± 19.47
RBS-R	15.50 ± 9.34	11.22 ± 9.31	21.47 ± 11.45	23.13 ± 14.19

SRS-2: Social Responsiveness Scale Second Edition; RBS-R: Repetitive Behavior Scale-Revised.

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
