# Peer review of "Effects of Mini-Basketball Training Program on Executive Functions and Core Symptoms among Preschool Children with Autism Spectrum Disorders"

_brainsci, 2020, doi:10.3390/brainsci10050263_

Round 1
Reviewer 1 Report
The present study implemented a basketball training program to improve executive functions and core symptoms among preschool-aged children with Autism Spectrum Disorder (ASD). Children were assigned to an experimental (basketball) group or a control group that did not receive the basketball intervention and maintained routine daily activities. The experimental (basketball) group was conducted over 12-weeks. Executive functioning and core symptoms were reported by the caregiver report. The results indicated that the experimental group demonstrated better executive functions and core symptoms compared to the control group. They also had improved social communication.
The study demonstrates novelty in its intervention design (basketball intervention) and could contribute greatly to the area of executive functioning and ASD. There are some issues that should be addressed before this manuscript can be in publishable form:
Abstract: Please clarify “all participants who had received their conventional rehabilitation program,” as this sentence is not grammatically correct and the meaning is not clear. After reading the paper in its entirety, I see that inhibition did not improve in the experimental group. Please correct this in the abstract.
Introduction: The authors provide clear rationale for their study and selected age-group. The authors rationalize that preschool age is a critical period for the development of EFs (line 76), but it would also be important for the authors to note that EFs also continue to develop into late adolescence and early adulthood. This section could be strengthened by the authors reviewing in greater detail the specific EFs that demonstrate growth during the preschool years. Regarding Line 78-80, the authors should link this text with implications for EFs. It is not clear to me what the authors mean by “mini-basketball” compared to regular basketball; please clarify.
Participants: I have concerns over the potential pre-existing differences in the experimental and control groups. All of the participants in the control group were recruited from Chuying, and all of the experimental group were recruited from Starssailor. Besides age and gender, were there any other demographic differences based on geographic location or the center-site? This should be addressed as limitation of the study design (in the Discussion section). Furthermore, there were 26 partcipants who did not complete their post-intervention assessment. What are the reasons for the attrition? Did the children who dropped out of the study differ in terms of demographic or participant characteristics (e.g., gender, age, core symptoms, family characteristics, etc.) than those who were able to successfully finish the 12-week intervention?
MBTP: Was there any data collected on the fidelity of the intervention? In other words, were all of the children able to complete all parts of the intervention, throughout all of the increasingly complex stages? Were there any children that took breaks during the 40 minute session? Were there any observed behavioral challenges? How about absences? Did any children miss one or more session during the 12-week duration? If this information was not collected, please address this as a limitation in the Discussion section.
Measurements: Please specify who completed these questionnaires (How many Mothers? Fathers?). Correct the “xxx” on line 201. Line 222: Please specify what parents were instructed to do during their play with the child during the training session. How much time did parents spend involved? Did the authors account for variability in how parents interacted with their children and how that might affect the efficacy of the MBTP? Regarding the CHEXI, please specify each of the four dimensions, since you describe them in the Results.
Statistical analyses: Spell out “effect sizes.
Results -- Executive functions: The authors note that the “2x2 mixed ANOVA revealed main effects for time,” but reported P>.05. If this main effect was not significant, then there was no main effect of time. Please reword to clarify.
Line 296: Please move this mention of Figure 3 earlier to clarify that it is about the Core symptoms only.
Discussion – Executive functions: Line 308 is not correct. Inhibitory control was not improved in the MBTP group, but rather improved for the control group. Please explain your interpretation for this finding.
Discussion – Core symptoms: Line 345: Please discuss how you think parental involvement might have contributed to the social communication findings. Include a discussion of the RRB findings as they relate to the intervention and previous literature.
Limitations – Please include in the limitations a lack of objective EF measure in the present study.
Reviewer 2 Report
The manuscript is focused on a very relevant and current issue. In addition, it is an interesting and a rigorous study. However, in some issues, it would be necessary to indicate more information. It can be useful in order to clarify some doubts that might arise in the future.
In order to improve the quality of the final publication, the following requirements and suggestions are indicated.
Line 48
Since these executive functions are dependent variables of the study (although it is not known until the results section), it would be necessary to define each of them.
Line 75
Notably, these above-mentioend physical….
It must be: Notably, these above-mentioned physical….
Line 104
Some relevant information about the characteristic of the participants is missing.
Which is the IQ of the participants? And their language level? And their severity level according to DSM-5?
To indicate this information is very necessary when we are working/research with people with ASD.
Line 116
The authors indicate that it is a pilot study. This could have been anticipated earlier (for example, in the abstract or in the Introduction: line 83: Hence, the purpose of the present PILOT study…)
Lines 118-119
…. 1) participants from the Chuying Child Development Center were chosen as the non-MBTP group (n=29);
The non-MBTP group could be called control group, as it is called in the rest of the manuscript. In this way, coherence will increase.
Lines 121-122
When you read this section of the article, information on the gender and age of the participants of each group is missing. Although it appears later (in Results section: Table 1), I think it would be clarifying to indicate it already here.
Line 176. Measurements Section
Information about psychometric properties of some of the assessment instruments is missing.
Line 191
It is necessary to indicate what are the 4 dimensions assesses with the CHEXI.
However, after, in Results, it appears only three. The authors do not indicate why one dimension (planning) is missing.
In addition, I have some questions:
- Why do the authors assess 4 dimensions if the authors of the CHEXI explain the following? However, factor analysis of CHEXI results from children in kindergarten was only able to identify two factors referred to as INHIBITION (inhibition and regulation subscales) and WORKING MEMORY (working memory and planning subscales).
- Why do the authors indicate that “A sum score of one hundred and twenty can be obtained…”? CHEXI does not offer an overall score.
Line 197
Which form SRS-2 of was used: school-age (ages 4-18 years), preschool (ages 2: 5-4: 5 years), both?
Authors do not mention the scales of SRS-2. This would imply a more real and complete description of the scale. (As it is indicated below, the authors do not use these scales. It would provide more information).
Line 201
Score is missing (…score of xxx that serves as…)
And why do the authors only are focused on the total score?
Line 203
Parentheses must be removed: …outcomes on (SRS-2) were….
Line 206
Authors do not mention the scales of RBS-R. This would imply a more real and complete description of the scale.
Line 228
It would be necessary to concrete what is a “fair remuneration”; i.e.: what /how much did the participants receive, exactly?
Line 279: 3.3. Core symptoms Section
Authors only inform about the total score of SRS-2 and not about the score of each of the scales of SRS-2. Why? To use the score of each scale will provide more information. It will be more interesting in order to determine the effects of MBTP.
The same issue occurs referring to RBS-R.
Lines 320-322
It would be very interesting and clarifier to include an example of the cognitive demands posed by the MBTP referring of each of executive functions assess in the study.
Lines 381-383
There is other methodology (observational methodology) that overcomes the limitations derived of the use of questionnaires. The authors should indicate it in their manuscript. There is some researches focused on assessment of preschool executive functions (Escolano-Pérez, Herrero-Nivela, Blanco-Villaseñor and Anguera, 2017) and assessment of executive functions in ASD children during and after an intervention carried out with observational methodology (Escolano-Pérez, Acero-Ferrero and Herrero-Nivela, 2019). It would be necessary to cite these researches in order to contribute to a better knowledge of how to carry out objective assessments of executive functions that overcome the limitations of the questionnaires.
Escolano-Pérez, E., Herrero-Nivela, M.L., Blanco-Villaseñor, A., and Anguera, M.T. (2017). Systematic observation: relevance of this approach in preschool executive function assessment and association with later academic skills. Front. Psychol. 8:2031. doi: 10.3389/fpsyg.2017.02031
Escolano-Pérez, E., Acero-Ferrero, M., Herrero-Nivela, M.L. (2019). Improvement of Planning Skills in Children With Autism Spectrum Disorder After an Educational Intervention: A Study From a Mixed Methods Approach. Front. Psychol. 10:2824. doi: 10.3389/fpsyg.2019.02824
Lines 383-385
Referring to SRS-2, the authors indicate that there is currently no other tool available which would allow for a more objective assessment of social communication impairments in preschool children diagnosed with ASD. Further support this statement would be necessary. Is it possible?
If there is currently no other tool available which would allow for a more objective assessment of these behaviors, there is other methodology that allow us it: the observational methodology
References:
31: It is necessary to indicate the abbreviated journal name.
32, 40: The journal names are missing
In general, it is necessary to check:
- the spaces before and after the signs = and <. It varies throughout the manuscript.
- P (p-value) must not be capital letter.
